# Effect of the Rho-Kinase/ROCK Signaling Pathway on Cytoskeleton Components

**DOI:** 10.3390/genes14020272

**Published:** 2023-01-20

**Authors:** Guangzhao Guan, Richard D. Cannon, Dawn E. Coates, Li Mei

**Affiliations:** 1Sir John Walsh Research Institute, Faculty of Dentistry, University of Otago, Dunedin 9016, New Zealand; 2Department of Oral Diagnostic and Surgical Sciences, Faculty of Dentistry, University of Otago, 310 Great King Street, Dunedin 9016, New Zealand; 3Department of Oral Sciences, Faculty of Dentistry, University of Otago, 310 Great King Street, Dunedin 9016, New Zealand

**Keywords:** cytoskeleton, Rho-kinase, ROCK, Rho-associated coiled-coil forming kinase

## Abstract

The mechanical properties of cells are important in tissue homeostasis and enable cell growth, division, migration and the epithelial-mesenchymal transition. Mechanical properties are determined to a large extent by the cytoskeleton. The cytoskeleton is a complex and dynamic network composed of microfilaments, intermediate filaments and microtubules. These cellular structures confer both cell shape and mechanical properties. The architecture of the networks formed by the cytoskeleton is regulated by several pathways, a key one being the Rho-kinase/ROCK signaling pathway. This review describes the role of ROCK (Rho-associated coiled-coil forming kinase) and how it mediates effects on the key components of the cytoskeleton that are critical for cell behaviour.

## 1. Introduction

The biomechanical properties of cells are inextricably linked to their intracellular structural components, such as the cytoskeleton [1]. Internal and external mechanical forces both act through the cytoskeleton to affect cellular mechanical properties, behaviour, cell spreading, movement, polarity and cytokinesis [2,3,4,5,6,7]. The cytoskeleton also plays essential biomechanical roles in connecting the plasma membrane of the cell, as well as internal membranes such as the endoplasmic reticulum, to the rest of the cell. The cytoskeletal network also regulates the diffusion of intracellular polymers that are larger than the network mesh size and possibly controls the permeation of water and small solutes through the cell [8]. Rho-associated coiled-coil forming kinase (ROCK) is considered to be a key regulator of the cytoskeleton and affects various important cellular functions such as cell shape, motility, secretion, proliferation and gene expression [9,10,11,12]. For example, various neural processes, including cell migration, axonal guidance, dendritic spine architecture, axonal regeneration and cell survival, are dependent on the ROCK signaling pathway [13]. The ROCK signaling pathway plays an important role in the reorganization of both microtubules and actin during the process formation of podocytes [14]. ROCK is associated with cancer progression and ROCK protein expression is higher in a number of cancer types such as primary stomach carcinoma, colon and bladder cancers and malignant melanoma [15]. A high level of ROCK protein expression correlates with poor overall survival in osteosarcoma and more aggressive behaviour in hepatocellular carcinomas [16,17,18]. This review is focused on the current knowledge of the regulation of the Rho-kinase/ROCK signaling pathway and the effects of ROCK on cytoskeletal components.

## 2. The Cytoskeleton

The main components of the cytoskeleton include microfilaments (specifically actin filaments), intermediate filaments and microtubules, which connect to each other, cellular organelles and the cell/nuclear membrane. They are biopolymers and organized into a three-dimensional network that resists deformation but can rearrange in response to internal or external forces and are thus responsible for maintaining the integrity of the cellular components. In general, there are four main functions of the cytoskeleton: organization of cellular contents; connection of cellular constituents to the external environment; mechanical resistance to deformation and generation of forces to move the cell or change its shape (Table 1) [2,19]. 

The three types of cytoskeletal components differ from each other, both chemically and physically. Generally, the most significant difference is their rigidity, which can be represented by the persistence length (*l*_P_) [7]. The persistence length is a mechanical property quantifying the bending stiffness of a polymer, which is defined as the distance over which the filament is bent by thermal forces and is proportional to the stiffness of the polymer [66]. If a polymer is considered to be a uniform cylinder, *l*_P_ = B_S_/k_B_T, where B_S_ is bending stiffness, k_B_ is the Boltzmann constant and T is the absolute temperature [67]. Filament stiffness, length, and geometry of cross-linking together determine the mechanical properties of cytoskeletal networks. The most common difference among the cytoskeletal filament types is their bending stiffness. The *l*_P_ of microfilaments is 10–17 μm, the *l*_P_ of intermediate filaments is less than 0.3–1.0 μm, and the *l*_P_ of microtubules is more than 1 mm [68]. This shows that microtubules are not only larger but also stiffer than microfilaments and intermediate filament.

## 3. Microfilaments

Microfilaments, or more specifically actin filaments, are semiflexible (*l*_P_ ~ 10–17 μm) polymers of actin (~375 amino acids with a molecular mass of ~42-kDa) folded into a U-shaped double helix structure. Each actin monomer has four subdomains and a bound adenine nucleotide, either ATP, ADP-Pi, or ADP, and a bound divalent cation, Mg^2+^ [67,69,70]. Microfilaments are ~7 nm in diameter and up to several micrometres in length [71]. Bending, torsional, and twist-bend coupling elasticities influence the microfilament’s mechanical properties [70]. The properties of microfilaments are determined by the strength and distribution of inter-subunit contacts [72]. Microfilaments can assemble into different structures, such as networks and bundles that are closely packed and crosslinked by trans-membrane adhesion proteins.

The mechanical properties of microfilament networks are determined by the physical properties of the individual actin molecules, the connections/interactions between actin monomers, and the three-dimensional geometry of the actin arrangement [8]. Microfilament networks appear to be the primary mechanical component of the cytoskeleton and are particularly abundant and dense near the inner surface of the plasma membrane, where they form a network with a variety of capping, binding, branching, and severing proteins. G-actin (individual globular actin) is capable of polymerisation and can bind either ATP or ADP. The pool of unpolymerized actin in normal cells is nearly exclusively made up of ATP-G-actin, with little to no ADP-G-actin in the pool [73]. Polymerization is not an energy-consuming process [74]. ATP is hydrolysed a long time after polymerization occurs [75]. F-actin is formed by polymerization and the addition of G-actin at the growing (+) end of the microfilament. The process of adding a G-actin unit at one end and removing it at the other (-) end is called treadmilling [76,77]. The actin cytoskeleton network is constantly assembling and disassembling by treadmilling, which can drive the formation of protrusive structures such as filopodia and lamellipodia. As a result, the network not only provides mechanical support, determines cell shape, but also drives cell locomotion by the extension of pseudopods, thereby enabling cellular migration, division, and particle engulfment [20,21,22]. The connection between the microfilaments and the trans-membrane adhesion proteins also facilitates communication between the intracellular and extracellular mechanical signals, which allows cells to detect and respond to both chemical and mechanical signals from their extracellular environment [78]. Microfilament networks are viscoelastic as they both store and dissipate mechanical energy [79]. The elasticity of the networks allows them to resist deformation like a simple spring. This resistance to deformation and ability to recover was found to be dependent on rho-kinase mediated contractility [80].

## 4. Intermediate Filaments

Intermediate filaments play an important role in cell mechanics, signaling and homeostasis. They belong to a superfamily of highly conserved, α-helix-rich fibrous proteins that are encoded by approximately 65–70 different genes [81,82,83,84]. They are more flexible biopolymers (*l*_P_ = 0.3–1.0 μm) than microfilaments and microtubules. Their diameter is from 8 to 12 nm, intermediate in size between microfilaments and microtubules [7]. Thus far, six major types of intermediate filament have been described [85], and their abundance varies between cell types. Examples of the different intermediate filaments are: acidic keratins (type I), basic keratins (type II), vimentin, glial fibrillary acidic protein (GFAP), peripherin and desmin (type III), neurofilaments (type IV), lamin (type V), and filensin and phakinin (type VI) [86]. Type I and II keratins are highly expressed in epithelial cells [29,32,87,88]. The type III protein vimentin is present mainly in cells of mesenchymal origin [41], whereas desmin, GFAP and peripherin are mainly found in muscle, glial cells, and neurons of the peripheral system, respectively [36,89]. Type IV proteins are found in many types of mature neurons [42,85]. Type V proteins are the nuclear lamins which form a filamentous support inside the inner nuclear membrane [36,47]. The type VI proteins, phakinin and filensin, are lens fibre cell specific intermediate filaments [52,56,90].

Intermediate filaments are vital determinants of intracellular organelle organization and form a delicate network in the cytoplasm, encapsulating the nucleus and radiating toward the plasma membrane [19]. They are the main proteins providing mechanical resistance to the cells, because of their flexibility, elasticity and extensibility [85]. Both in vitro and in vivo studies have shown that intermediate filaments are important contributors to the elasticity and tensile strength of cells [91,92,93]. Indeed, intermediate filaments have been shown to determine cell stiffness in keratinocytes [93]. Indirect perturbation of cytoplasmic intermediate filaments has detrimental effects on cell stiffness. Research has shown that lipids such as sphingosylphosphorylcholine, induce perinuclear rearrangement of intermediate filaments leading to a significant increase in cellular elasticity, which could facilitate metastasis [94].

## 5. Microtubules

Microtubules are cylindrical polymers of α and β-tubulin dimers. There are multiple genes encoding tubulin protein isotypes [95,96]. According to The HUGO Gene Nomenclature Committee, there are 24 tubulin genes (10 alpha tubulins, 10 beta tubulins, 1 delta, 1 epsilon and 2 gamma tubulins) and 3 pseudogenes [97]. These tubulin proteins polymerize into protofilaments (α–β)n, and arrange in a spiral to form a hollow tube with a 25 nm diameter [67,98]. In addition, γ-tubulin which is located at the centrosome, has an important role in initiating microtubule assembly. Microtubules have an *l*_P_ > 1 mm and are stiffer than microfilaments and intermediate filaments. They are relatively brittle and can be fractured more easily than intermediate filaments [68]. The published flexural rigidity ranges from 1 × 10^−24^ to 32 × 10^−24^ Nm^2^ [99]. Microtubules form a dynamic scaffold for the cells and are involved in a variety of functions, including the transportation of intracellular vesicles and organelles throughout the cell during normal physiological processes, force generation (by polymerization, depolymerization, or interactions with motor proteins), formation of intracellular transport platforms (for anchoring, signaling and force-coupling roles), segregation of chromosomes, assembly of the mitotic spindle in dividing cells, axon extension in neurons, directional cell migration and differentiation, maintaining cell stiffness/shape, and regulation of cell morphology and cell mechanics [57,100,101]. Microtubules undergo constant cycles of polymerization and depolymerization utilizing a guanosine triphosphate (GTP) dependent process (dynamic instability). The polymerization of microtubules begins with the formation of 13 linear protofilaments which are composed of head to tail arrays of tubulin dimers. Similar to actin, microtubules have two distinct ends (plus and minus ends). The unprotected minus ends are unstable and often require stabilizer proteins (calmodulin-regulated spectrin-associated proteins or Partonins) to prevent depolymerization [102]. On the other hand, microtubules elongate by the addition of tubulin subunits to the plus end [103]. They grow by αβ-tubulin heterodimers bound to GTP binding to the growing end of the microtubule. The GTP bound to the αβ-tubulin heterodimers can be hydrolyzed to GDP during or immediately after polymerization [104]. This hydrolysis decreases the binding affinity of tubulin polymers for adjacent molecules, thereby favouring depolymerization and resulting in the dynamic behaviour of microtubules (alternation between cycles of growth and shrinkage) [105]. Microtubules can therefore undergo periods of growth and disassembly, which means that microtubules have half-lives of only several minutes within the cell [106]. This is particularly important for the remodelling of the cytoskeleton that occurs during mitosis, and for force generation and signaling [107]. In order to control this dynamic instability, many organisms have developed proteins that perturb microtubule dynamics, several of which are in use as cancer chemotherapeutics and anti-inflammatory drugs [105]. For example, the anti-cancer drug paclitaxel (Taxol) stabilizes microtubules and prevents their disassembly, promoting mitotic arrest and cell death [108]. In epithelial cells, microtubules are responsible for the spatial organization of the secretory and endocytic apparatuses, the facilitation of exocytic and post-endocytic protein transportation, participation in protein sorting and apico-basolateral polarity [109].

In addition to these three main cytoskeleton components, many other proteins are associated with the cytoskeleton, such as spectrin, kinesin, plectin, emerin, Sad1 and UNC-84 proteins, Klarsicht, ANC-1, and Syne homology proteins, and they are important for cellular mechanics. [19]. Together, the cellular mechanical properties are determined by complex interactions involving cytoskeletal proteins and polymers. There are several pathways that lead to major rearrangements of the cytoskeletal structures, namely: Rho-kinase/ROCK; PI3K/AKT; β-Catenin-independent Wnt (Wnt ligands stabilize β-catenin; β-catenin helps link cadherin adhesion molecules to cytoskeleton); glycogen synthase kinase-3 beta (Gsk-3β); Mitogen-activated protein kinase (MAPK); and cyclic GMP-dependent protein kinase [110,111,112,113,114,115,116,117,118,119] (Table 2). 

Other kinases, such as PI4P and protein kinase C, are also important in regulating interactions of the cell membrane proteins and protein scaffolds involved in vesicle budding, and cytoskeletal organization [197,198]. In particular, ROCK has been considered as a key regulator of the cytoskeleton as it mediates various important cellular functions such as cell shape, motility, secretion, proliferation, and gene expression [199].

## 6. Rho-Associated Kinase/ROCK Pathway and Associated Genes 

### 6.1. Rho GTPases

Rho GTPases are key regulators of several cellular processes, including cytoskeletal shape, gene expression, cell cycle progression, cell polarity and cell migration [200,201,202]. Rho GTPases are a sub-family within the Ras superfamily of G-proteins and comprise at least 20 members, including Rho (A, B, C isoforms), Rac (1, 2, 3 isoforms), Cdc42 (Cdc42Hs, G25K isoforms), Rnd1/Rho6, Rnd2/Rho7, Rnd3/RhoE, RhoD, RhoG, TC10 and TTF [203,204,205]. In particular, RhoA, RhoB and RhoC, have similar effectors and modes of action, which impact many cellular processes [206]. They serve as gate control molecules by cycling between a GTP-bound active and GDP-bound inactive states-facilitated by Rho GTPase-specific guanine nucleotide exchange factors (GEFs) [207,208,209]. Subsequently, Rho proteins regulate downstream pathways which affect cell migration, adhesion, proliferation, apoptosis and the cell cycle via induction of contractile fibre bundles such as microfilaments, modulating microtubules and regulating intermediate filament turnover [203,204,210,211]. Rho GTPases interact with at least 30 effector proteins and initiate downstream signaling [212,213]. The effectors of three members of the family (RhoA, Rac1 and Cdc42) have been well studied. RhoA regulates the contraction of moving cells, whereas Rac1 and Cdc42 mediate the formation of lamellipodia and filopodia, respectively [214]. Rho GTPase effectors can be classified into different groups according to their functions, such as scaffold proteins, kinases, actin-binding proteins, formin-like molecules, and phospholipases [215]. For example, ROCK1,2 (serine/threonine kinases) and Dia1,2 (scaffold proteins) are important RhoA effectors that regulate cytoskeleton nucleation and polymerization [206]. Wiskott–Aldrich syndrome protein (WASP; a scaffold protein) and neural-WASP (scaffold protein N-Wasp) are Cdc42 effectors. N-Wasp mediates actin polymerization via the Arp2/3 complex [216]. Formin-homology-domain-containing protein (formin-like molecule Fhod1) is one of the Rac1 effectors that regulates actin cytoskeleton organization and gene transcription [217]. p21-activated (serine/threonine) kinase is the effector for both Rac1 and Cdc42 [218], it regulates cell shape and polarity through phosphorylation of multiple cytoskeletal proteins.

### 6.2. ROCK

Rho-associated kinase ROCK is one of the most important effectors downstream of Rho GTPase [219] (Figure 1). Human ROCKs consist of two isoforms, ROCK 1 and ROCK 2. ROCK 1 is located on chromosome 18 (18q11.1, human), and its mRNA is highly expressed in bone marrow and adipose cells (https://www.ncbi.nlm.nih.gov/gene/6093#gene-expression, accessed on 21 December 2022) whereas ROCK2 is located on chromosome 2 (2p25.1, human) and its mRNA is expressed abundantly in adipose tissue and the colon (https://www.ncbi.nlm.nih.gov/gene/9475, accessed on 21 December 2022) [220,221]. ROCK proteins are activated by binding Rho-GTPase which then, directly and indirectly, regulate the cytoskeleton. ROCK proteins induce a wide range of cellular responses that involve microfilaments, intermediate filaments and microtubules (Figure 2). Their downstream targets are membrane distal, including LIM kinases, myosin phosphatase target subunit of myosin light chain phosphatase, myosin light chain (MLC), collapsing response mediator protein, and ezrin-radixin-moesin (ERM) proteins. For example, ROCK controls the organization and stabilization of actin filaments by phosphorylating a number of proteins, such as MLC [222], LIM1/2 [223], myosin phosphatase target subunit 1 (MYPT1) [224], ERM [225], adducin [226], calponin [227], myristoylated alanine-rich C kinase substrate (MARCKS) [228], elongation factor-1 alpha (EF1α) [229], troponin I/T [230] and profilin [231]. ROCK phosphorylates MLC either directly or inactivates MLC phosphatase, resulting in the induction of actin-myosin contractility [232]. ROCK also activates LIM-Kinase by phosphorylation and LIM-Kinase in turn phosphorylates cofilin, which inhibits cofilin’s actin depolymerization activity, resulting in the stabilization of the actin cytoskeleton [223] (Figure 2). Both the ROCK/MYPT1/MLC and ROCK/LIM kinases/cofilin pathways are key elements in stress fibre assembly and cell adhesion [233]. Moreover, although ROCK1 and 2 have high sequence identity, they differ functionally, especially in relation to actin regulation (Figure 1). ROCK 1 is thought to be involved in destabilizing actin via regulating MLC and actin-myosin contraction, whereas ROCK 2 acts via cofilin to stabilize the actin cytoskeleton [233]. However, Wang et al., found that ROCK 1 phosphorylates LIM-kinase to inactivate cofilin, so regulating cell adhesion and invasion, but not ROCK 2 does not [234]. Also, Rochelle et al., demonstrated that ROCK 1 mediated amoeboid motility via destrin but not via cofilin [235].

ROCK proteins regulate different types of intermediate filaments via phosphorylation. For example, ROCK proteins can breakdown and separate the intermediate filaments via phosphorylation. It also upregulates the cell stiffness by interacting with keratin intermediate filaments [130,236,237,238,239]. Interestingly, activation of ROCK induces the deformation of intermediate filaments (Figure 2) with an instant release of inactive ROCK. The released ROCK is transported to the periphery of the cell and reactivated by Rho-GTP, it can then accelerate the phosphorylation and disassembly of intermediate filaments [240]. In addition, ROCK regulates microtubules by phosphorylation of several microtubule-associated proteins such as MAP2/Tau [241], collapsin response mediator protein 2 (CRMP2) [242] and Doublecortin [243] (Figure 2). ROCK-induced phosphorylation of MAP2/Tau leads to destabilization of the microtubules [12,244]. CRMPs stabilize microtubules however this is inhibited by phosphorylation via ROCK [245]. Doublecortin is a microtubule-associated protein. Phosphorylation of Doublecortin via ROCK inhibits microtubule bundling [243]. In addition, ROCK regulates microtubule acetylation via phosphorylation of the tubulin polymerization promoting protein 1 (TPPP1/p25), resulting in a decreased cellular level of acetylated tubulin and increased cell migration [246].

There are six genes that either enhance (disabled 2 (Dab2), synaptopodin 2 (Synpo2) and thymus cell antigen 1, theta (Thy1)) or inhibit (cysteine and histidine-rich domain (CHORD)-containing, zinc-binding protein 1 (Chordc1), heart of glass (Heg1), and Ras interacting protein 1 (Rasip1)) ROCK activity. For example, Dab2 (on chromosome 15), which plays an important role in cell proliferation and differentiation, is capable of upregulating RhoA/ROCK signaling [247]. SYNPO2 (known as myopodin), encodes an actin-binding protein, and has been characterized as a tumour suppressor which positively regulates ROCK [248]. It may promote cell migration through activation of the ROCK signaling pathway via increasing levels of Rho-GTP [249]. Thy1 (on chromosome 9) triggers the actin cytoskeleton remodelling via the Thy-1-CBP-Csk-Src-RhoA-ROCK axis [250]. However, overexpressed Morgana/chp-1, encoded by the CHORDC1 gene (on chromosome 9), negatively regulates Rho, and binds then inhibits ROCK 1 and 2 [251]. Rasip1 (on chromosome 7) increases the activity of Cdc42 and suppresses actomyosin contractility via inhibition of RhoA [252]. Silencing HEG1 (on chromosome 16) expression by siRNA or shRNA increases phosphorylation of MLC and formation of stress fibres indicating increased ROCK signaling [253]. Together with these regulating genes, the ROCK signaling pathway plays an important role in cytoskeleton and cellular mechanics. 

ROCK inhibitors act to suppress the ROCK pathway through multiple mechanisms. They can relax the trabecular meshwork through inhibiting the actin cytoskeleton contractile tone. They can induce beta-catenin nuclear translocation and inhibit cell migration or block ATP-dependent phosphorylation therefore inhibiting ROCK1 and ROCK2 and can inhibit Rho GTPases from binding to ROCK [233,254,255]. More than 170 chemicals are capable of inhibiting ROCK, either selectively or non-selectively [256]. Most of them are ATP-competitive kinase inhibitors [257,258]. Structurally, they can be classified into isoquinoline/isoquinolinone, indazole, pyridine, pyrimidine, pyrrolopyridine, pyrazole, benzimidazole, benzothiazole, benzathiophene, benzamide, phthalazinone, aminofurazan, quinazoline, and boron derivatives [256]. Several ROCK inhibitors have been used in clinical trials for many therapeutic indications, such as ophthalmology, cardiovascular disease, anti-erectile dysfunction, and cancers. For example, both Ripasudil and Netarsudil have been proven to be of benefit in the treatment of glaucoma [259]. Fasudil, which is a potent vasodilator, has been approved for the treatment of cerebral vasospasm in Japan and China since 1995 [260]. Y-27632 improved ventricular hypertrophy, fibrosis, and function in a rat model [261,262]. Daily administration of Y-27632 also improved erectile responses in a rat model [263,264]. The use of ROCK inhibitors in cancer, such as gastric, bone, lung, breast, renal, liver and prostate cancers, has been widely characterised [265,266,267,268,269,270,271]. For example, the selective ROCK inhibitor RKI-1447 was shown to have significant anti-invasive and anti-tumour effects [272]. AT13148, an inhibitor of ROCK and AKT kinases, has been found to have antimetastatic and antiproliferative activity [273]. Despite the potential of ROCK inhibitors, the number of clinical trials for human cancer is still limited.

Although ROCK 1 and ROCK 2 are highly homologous kinases and are both involved in the Rho/ROCK signaling pathway, their target substrates and physiological/pathological activities differ. Unfortunately, most ROCK inhibitors cannot selectively target one or other of the two isoforms; such inhibitors would be very useful. Dimerization is required for the RhoE activation of ROCK 1 [274], and although the homology between the kinase domains of ROCK 1 and ROCK 2 is ~90%, the homology between the N-terminal dimerization domains [275] of the two proteins is only ~60% [276]. Therefore dimerization-disrupting peptides could be developed and used to selectively target the dimerization domains of the two isoforms [276]. Autophosphorylation of ROCK 1 at Ser1333 and ROCK 2 at Ser1366 is required for activation of the kinases [277]. It has been shown that the interaction between Thr405 in the hydrophobic motif of ROCK 2 and Asp39 in the N-terminal extension was essential for both kinase activation and dimerization [278]. This provides further support for the development of peptides that prevent dimerization and kinase activity of ROCK isoforms.

## 7. Conclusions

The cytoskeleton gives cells their shape, structure and mechanical support and plays an essential role in cellular mechanics. ROCK is an effector of the small Rho GTPase and when activated influences many cellular functions and processes by regulating different elements of the cytoskeleton. Many cellular and physiological functions are mediated by ROCK and its activity is often elevated in some disorders, making them good targets for the development of new drugs. ROCK inhibitions have been found to effectively manage several diseases in humans and animal models. Progress has been made towards understanding how non-selective ROCK inhibitors work for several diseases and conditions and efforts should be made to develop ROCK inhibitors with improved specificity and sensitivity.

## Figures and Tables

**Figure 1 genes-14-00272-f001:**
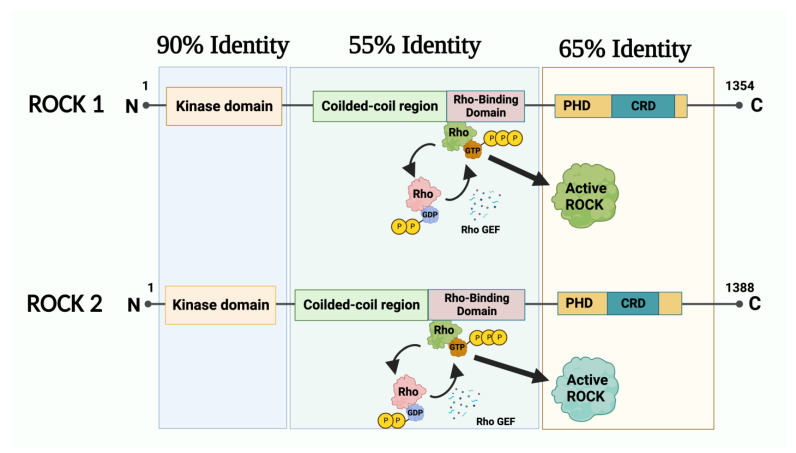
Molecular structure of ROCK 1 and ROCK 2. They are similar and both contain a kinase domain, coiled-coil region, rho-binding domain, putative pleckstrin homology domain (PHD) and cysteine-rich domain (CRD). Rho GEF = Rho guanine nucleotide exchange factor. Created with BioRender.com.

**Figure 2 genes-14-00272-f002:**
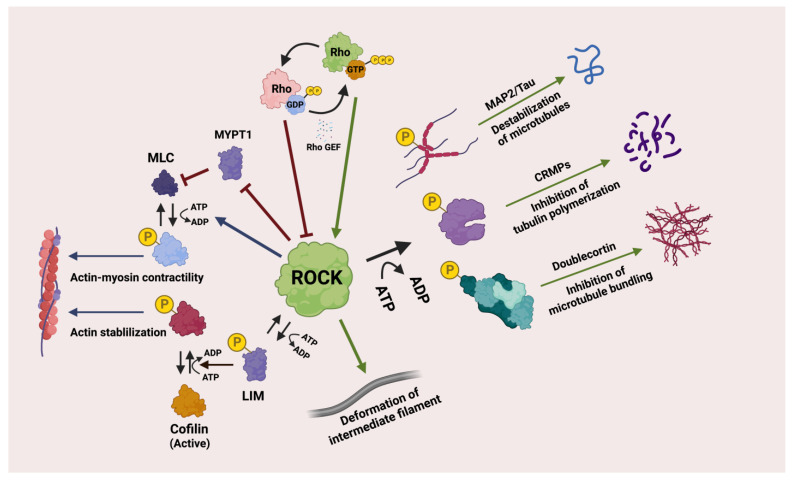
The functions of ROCK. Rho is activated by Rho guanine nucleotide exchange factor (GEF). Activated GTP-bound Rho stimulates ROCK. Activated ROCK activates or inhibits downstream targets by phosphorylation. Activated ROCK regulates microfilaments via the ROCK/MYPT1/MLC and ROCK/LIM kinases/cofilin pathways. ROCK also regulates intermediate filaments by phosphorylation. ROCK regulates microtubules by phosphorylation of several microtubule-associated proteins such as MAP2/Tau, collapsin response mediator protein 2 (CRMP2) and Doublecortin. Created with BioRender.com.

**Table 1 genes-14-00272-t001:** Cytoskeleton components and their functions.

Cytoskeleton Component	Functions	References
**Microfilament**		
Actin	Generation of forces in cellular contraction, endo- and exocytosis, secretion, vesicle transfer, cell division and cell integrity	Jiang et al., 2021 [20]; Falahzadeh et al., 2015 [21]; Gordon-Alonso et al., 2010 [22]; Blessing et al., 2004 [23]; Halpain 2003 [24]; Lanier and Gertler, 2000 [25]; Pollard et al., 2000 [26]; Schmidt and Hall, 1998 [27]; Bretscher, 1993 [28].
**Intermediate filament**		
Type I (Acidic Keratins)	Expressed in epithelial cells, form the structural framework of cells and contribute to mechanical resilience	Gül et al., 2022 [29]; Jacob et al., 2018 [30]; Saitoh et al., 2016 [31]; Schweizer et al., 2006 [32].
Type II (Basic Keratins)	Expressed in epithelial cells, they also maintain the stability of the cell nucleus and mechanical stability of the whole cell	Honda et al., 2014 [33]; Infante et al., 2011 [34]; Moll et al., 2008 [35]; Bowden et al., 1984 [36].
Type III (Vimentin)	Regulate cell mechanics and coordinate mechanosensing, transduction, signaling pathways, motility and inflammatory responses	Ridge et al., 2022 [37]; Yue et al., 2016 [38]; Kidd et al., 2014 [39]; Menko et al., 2014 [40]; Herrmann et al., 2007 [41]; Wang et al., 2006 [42]
Type IV (Neurofilaments)	Provide structural support for axons and regulate axon radial growth	Didonna and Opal, 2019 [43]; Yuan andNixon, 2016 [44]; Yuan et al., 2012 [45]; Lin and Schlaepfer, 2006 [46]; Hoffman1988 [47]
Type V (Nuclear Lamins)	Attach chromatin domains to the nuclear periphery and localize some nuclear membrane proteins	Khadija et al., 2015 [48]; Carmosino et al., 2014 [49]; Burke and Stewart, 2013 [50]; Lopez-Soler et al., 2001 [51]; Parnaik, 2008 [52]
Type VI (Phakinin and Filensin)	Lens fiber and maintenance of lens transparency	Oka et al., 2008 [53]; Pittenger et al., 2007 [54]; Blankenship et al., 2001 [55]; Goulielmos et al., 1996 [56]; Georgatos et al., 1994 [57]
**Microtubule**		
Tubulin	Maintain the structure of the cell, transport secretory vesicles, organelles and macromolecular assemblies and assembly of mitotic spindle	Hlavaty and Lechler, 2021 [58]; Chang and Gu, 2020 [59]; Matis 2020 [60]; Logan and Menko, 2019 [61]; Cirillo et al., 2017 [62]; Ganguly et al., 2012 [63]; van der Vaart et al., 2009 [64]; Kulić et al., 2008 [65]

**Table 2 genes-14-00272-t002:** Pathways that regulate the cytoskeleton.

Pathway	Function	Cytoskeleton Components Affected	Authors, Year
**Rho-kinase/ROCK**	Major signaling pathway that regulates the cytoskeleton and cell polarity	Actin	Shi and Wei, 2022 [120]; Tang et al., 2018 [121]; Schofield and Bernard, 2013 [122]; Sit and Manser, 2011 [123]; Sarasa-Renedo et al., 2006 [124]; Woods et al., 2005 [125]; McBeath et al., 2004 [10]; Da Silva et al., 2003 [126]; Amano et al., 2000 [127]; Maekawa et al., 1999 [11]
		Intermediate filaments	Tang et al., 2018 [121]; Yang et al., 2017 [128]; Lei et al., 2013 [129]; Schofield et al., 2013 [122]; Amano et al., 2010 [12]; Hirose et al., 1998 [130]
		Microtubules	Becker et al., 2022 [131]; Schofield et al., 2013 [122]; Heng et al., 2012 [132]; Fonseca et al., 2010 [133]; Takesono et al., 2010 [134]; Gao et al., 2004 [14]
**PI3K/AKT**	Roles in the assembly of actin filaments, polymerization of microtubules and intermediate filaments	Actin	Han et al., 2020 [135]; Lien et al., 2017 [136]; Kakinuma et al., 2008 [137]; Qian et al., 2005 [138]; Qian et al., 2004 [139]; Krasilnikov 2000 [140]
		Intermediate filaments	Deng et al., 2022 [117]; Roux et al., 2017 [141]; Wang et al., 2012 [142]; Kong et al., 2012 [143]; Tseng et al., 2011 [144]
		Microtubules	Chakrabarty et al., 2019 [145]; Fu et al., 2017 [146]; Kitagishi et al., 2014 [147]; Onishi et al., 2007 [148]; Fujiwara et al., 2007 [149];
**Wnt/β-catenin**	Cell proliferation, differentiation, survival and adhesion (cytoskeleton reorganization) and regulation of microtubule stability. Wnt ligands stabilize β-catenin whereas β-catenin helps link cadherin adhesion molecules to cytoskeleton	Actin	Zhang et al., 2022 [150]; Roarty et al., 2017 [115]; Galli et al., 2012 [151]; Lai et al., 2009 [110]; James et al., 2008 [114]
		Intermediate filaments	Lehmann et al., 2020 [152]; Tian et al., 2019 [153]; Bermeo et al., 2015 [154]; Prasad et al., 2008 [155]; Bierie et al., 2003 [156]
		Microtubule	Puri et al., 2021 [157]; Ou et al., 2020 [158]; Huang et al., 2007 [159]; Ciani et al., 2004 [160]; Peifer et al., 2000 [161]
**Gsk-3β**	Inhibition of Wnt signaling pathway. Responsible for actin branching, regulation of intermediate filaments and controlling microtubule dynamics	Actin	Hajka et al., 2021 [162]; Yoshino et al., 2015 [163]; Watanabe et al., 2009 [164]; Sun et al., 2009 [165]; Vaidya et al., 2006 [166]
		Intermediate filaments	Sen et al., 2022 [167]; Lee et al., 2012 [168]; Kim et al., 2012 [169]; Sasaki et al., 2002 [170]; Guidato et al., 1996 [171]
		Microtubules	Sen et al., 2022 [167]; Hajka et al., 2021 [162]; Beurel et al., 2015 [116]; Watanabe et al., 2009 [164]; Fumoto et al., 2006 [172]; Grimes and Jope, 2001 [173]
**MAPK**	Cytoskeleton remodelling, downregulation of vimentin and affects the polymerization and stability of microtubules	Actin	Joe et al., 2022 [174]; Hoffman et al., 2017 [111]; Yang et al., 2007 [175]; Fujiwara et al., 2005 [176]; Paliga et al., 2005 [177]
		Intermediate filaments	Tania et al., 2014 [178]; Wang et al., 2021 [179]; Wöll et al., 2007 [180]; Schechter et al., 1998 [181]; Cheng and Lai, 1998 [182]
		Microtubules	Li et al., 2015 [183]; Hu et al., 2010 [184]; Lee et al., 2007 [185]; Fan and Chambers, 2001 [186]; Reszka et al., 1995 [187];
**cGMP kinase**	Inhibition of actin cytoskeleton organization, phosphorylation of vimentin and modulating microtubules and their associated proteins	Actin	Zou et al., 2018 [113]; Butt et al., 2003 [188]; Butt et al., 2001 [189]; Sandau et al., 2001 [190]; Sauzeau et al., 2000 [112]
		Intermediate filaments	Pryzwansky et al., 1995 [191]; MacMillan-Crow and Lincoln, 1994 [192]; Wyatt et al., 1993 [193]; Wyatt et al., 1991 [194]
		Microtubules	Xia et al., 2013 [195]; Gong et al., 2011 [196]

Glycogen synthase kinase-3 beta (Gsk-3β), Mitogen-activated protein kinase (MAPK), Cyclic GMP-dependent protein kinase (cGMP kinase).

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
