# Peer review of "Effect of the Rho-Kinase/ROCK Signaling Pathway on Cytoskeleton Components"

_genes, 2023, doi:10.3390/genes14020272_

Round 1

Reviewer 1 Report

No major comment.

The topic is too general. It is suggested to include the role of Rho-Kinase/ROCK in cells with high dynamic cytoskeletons like neurons, podocytes, cancer cells, ….

Please add a section about ROCK inhibitors with updated references.

The article would require being revised significantly to improve the grammar, typos, and structures.

Author Response

1st Jan 2023

Genes

Re: Manuscript ID: genes-2140180-ori

We are grateful for the reviewers’ comments. We have revised the manuscript based on the comments. The major changes have been tracked in red in this revision.

Comments from Reviewer 1:

The topic is too general. It is suggested to include the role of Rho-Kinase/ROCK in cells with high dynamic cytoskeletons like neurons, podocytes, cancer cells, ….

Please add a section about ROCK inhibitors with updated references.

The article would require being revised significantly to improve the grammar, typos, and structures.

Response: We have included a description of the role of Rho-Kinase/ROCK in cells with dynamic cytoskeletons like neurons, podocytes, cancer cells (page 1, line 34-42), added a section about ROCK inhibitors (Page 6, line 274-298), and improved the grammar, spelling, and structure.

Reviewer 2 Report

The review by Guangzhao Guan and coll., “Effect of Rho-kinase/ROCK signaling pathway on cytoskeleton components” aims to cover the relationships between ROCK and cytoskeleton. This could be an interesting approach since a lot of data concerning ROCK are available but should need classification. In fact, the review focuses on cytoskeleton, in an unoriginal way, and is very short on its purpose, ROCK. One would prefer a clear development on ROCK isoforms, and a classification of activated or inactivated pathways, the type of cells in which the results were obtained and a comprehensive parallel with pathophysiology.

There are also a lot of concerns in the text. First, there are hyphens everywhere that avoid fluid reading.

L75: The authors claim that ATP provides energy for actin polymerization. This is not true. Polymerization is not energy consuming. ATP is hydrolyzed a long time after polymerization occurs (and it’s the same for GTP and tubulin).

L121: For tubulin genes, the authors refer to the Nature review in which the Hugo gene nomenclature is falsely cited. Going to this nomenclature gives 10 alpha tubulins, 10 beta tubulins, 1 delta (not cited), 1 epsilon (not cited) and 2 gamma tubulins (24 genes and 3 pseudogenes). Please modify.

L162: lamins are intermediate filaments, then belong to the cytoskeleton.

L188: the authors describe very shortly Rho GTPases effectors and conclude that their functions are not yet fully understood. The reference to this low understanding is a 1997 paper by Watanabe et al., which is not appropriated for such affirmation. Also, for some of effectors, a better understanding is now obtained and could have been developed.

L208 and 211: MLC is here called MLC2, which is wrong. Maybe it is a contraction between MLC and myosin II? Please modify.

Here also, for the different pathways activated by ROCK 1 or ROCK2, there is more literature to cite. One would like a development of this part.

L215: The sentence is impossible to understand. A reference to cytokinesis seems underlying, but it needs to be clarified.

L 246-247: the conclusion is devoid of any thought.

This reviewer does not understand the utility of table I and II. There are no correspondence to them in the text and they do not enhance the understanding of the subject (for table I) or are totally out of the focus of this review (table II).

Author Response

1st Jan 2023

Genes

Re: Manuscript ID: genes-2140180-ori

We are grateful for the reviewers’ comments. We have revised the manuscript based on the comments. The major changes have been tracked in red in this revision.

Comments from Reviewer 2:

The review by Guangzhao Guan and coll., “Effect of Rho-kinase/ROCK signaling pathway on cytoskeleton components” aims to cover the relationships between ROCK and cytoskeleton. This could be an interesting approach since a lot of data concerning ROCK are available but should need classification. In fact, the review focuses on cytoskeleton, in an unoriginal way, and is very short on its purpose, ROCK. One would prefer a clear development on ROCK isoforms, and a classification of activated or inactivated pathways, the type of cells in which the results were obtained and a comprehensive parallel with pathophysiology.

There are also a lot of concerns in the text. First, there are hyphens everywhere that avoid fluid reading.

Response: Thanks, and we have improved the structure and corrected the grammar mistakes.

L75: The authors claim that ATP provides energy for actin polymerization. This is not true. Polymerization is not energy consuming. ATP is hydrolyzed a long time after polymerization occurs (and it’s the same for GTP and tubulin).

Response: We have updated and corrected the manuscript with new references (Page 2, line 84-88).

“G-actin (globular actin) is capable of polymerisation and can bind either ATP or ADP. The pool of unpolymerized actin in normal cells is nearly exclusively made up of ATP-G-actin, with little to no ADP-G-actin in the pool [27]. Polymerization is not an energy-consuming process [28]. ATP is hydrolysed a long time after polymerization occurs [29].”

L121: For tubulin genes, the authors refer to the Nature review in which the Hugo gene nomenclature is falsely cited. Going to this nomenclature gives 10 alpha tubulins, 10 beta tubulins, 1 delta (not cited), 1 epsilon (not cited) and 2 gamma tubulins (24 genes and 3 pseudogenes). Please modify.

Response: We have updated the manuscript with new references.(Page 3, line 134-136)

“According to The HUGO Gene Nomenclature Committee, there are 24 tubulin genes (10 alpha tubulins, 10 beta tubulins, 1 delta, 1 epsilon and 2 gamma tubulins) and 3 pseudogenes [56].”

L162: lamins are intermediate filaments, then belong to the cytoskeleton.

Response: We have removed reference to lamins from the microtubule section of the manuscript. (Page 4, line 175)

L188: the authors describe very shortly Rho GTPases effectors and conclude that their functions are not yet fully understood. The reference to this low understanding is a 1997 paper by Watanabe et al., which is not appropriated for such affirmation. Also, for some of effectors, a better understanding is now obtained and could have been developed.

Response: We have updated the manuscript accordingly. (Page 5, line 203-211)

L208 and 211: MLC is here called MLC2, which is wrong. Maybe it is a contraction between MLC and myosin II? Please modify.

Response: We have updated the manuscript accordingly.(Page 5, line 232)

Here also, for the different pathways activated by ROCK 1 or ROCK2, there is more literature to cite. One would like a development of this part.

L215: The sentence is impossible to understand. A reference to cytokinesis seems underlying, but it needs to be clarified.

Response: We have updated the manuscript accordingly.(Page 5, line 240-243)

L 246-247: the conclusion is devoid of any thought.

Response: We have updated the manuscript accordingly.(Page 7, line 303-310)

This reviewer does not understand the utility of table I and II. There are no correspondence to them in the text and they do not enhance the understanding of the subject (for table I) or are totally out of the focus of this review (table II).

Response: Thank you for your comments. Table 1 was to illustrate the different functions of the cytoskeleton. Table 2 was to show different pathways that regulate the cytoskeleton.

Reviewer 3 Report

G. Guan and co-authors propose a review on the roles of the rho-kinase (ROCK) in the control of the cytoskeleton organization, and its regulation. The manuscript is well written and illustrated, and recapitulates main informations in the field. However some complements and minor corrections are required:

- The role of ROCK in cell survival should be indicated.

- The respective roles of beta catenin and alterne Wnt pathway in the regulation of the cytoskeleton should be indicated in the text and Table 2.

-Beside the role of PI3K in the regulation of the cytoskeleton, it should be important to add informations on the roles played by other kinases such as PI4K and PKC.

- Inappropriate spaces cutting words (carriage return) should be checked.

- The abbreviation LP must be indicated in parentheses.

- Line 125: please discard "the" before stiffer.

- Lines 209 and 210, including reference to figure 1, should be placed at the beginning of the paragraph "ROCK".

- Line 204: Please change for "or inactivates MLC...".

- Lines 215-216: The sentence should be rewritten to clarify.

- Line 247: Please add a S to play.

- Figure 2: Please correct for "inhibition of tubulin polymerization" and add a S to "function" in the title. Discard "The" before Rho and ROCK. Add a S to "microfilament". 

Author Response

1st Jan 2023

Genes

Re: Manuscript ID: genes-2140180-ori

We are grateful for the reviewers’ comments. We have revised the manuscript based on the comments. The major changes have been tracked in red in this revision.

Comments from Reviewer 3:

  1. Guan and co-authors propose a review on the roles of the rho-kinase (ROCK) in the control of the cytoskeleton organization, and its regulation. The manuscript is well written and illustrated, and recapitulates main informations in the field. However some complements and minor corrections are required:

- The role of ROCK in cell survival should be indicated.

Response: We have added the role of ROCK in cell survival (Page 6, line 291-293).

- The respective roles of beta catenin and alterne Wnt pathway in the regulation of the cytoskeleton should be indicated in the text and Table 2.

Response: We have updated the manuscript accordingly (Page 4, line 180-181) and table 2. “Wnt ligands stabilize β-catenin. β-catenin helps link cadherin adhesion molecules to cytoskeletal actin filaments (79, 80).”

-Beside the role of PI3K in the regulation of the cytoskeleton, it should be important to add informations on the roles played by other kinases such as PI4K and PKC.

Response: We have updated the manuscript accordingly. (Page 4, line 183-185)

- Inappropriate spaces cutting words (carriage return) should be checked.

Response: Thanks, and we have corrected all the structural and grammar mistakes.

- The abbreviation LP must be indicated in parentheses.

Response: We have updated the manuscript accordingly.(Page 2, line 56)

- Line 125: please discard "the" before stiffer.

Response: We have updated the manuscript accordingly.(Page 2, line 66)

- Lines 209 and 210, including reference to figure 1, should be placed at the beginning of the paragraph "ROCK".

Response: We have updated the manuscript accordingly.(Page 5, line 214)

- Line 204: Please change for "or inactivates MLC...".

Response: We have updated the manuscript accordingly.(Page 5, line 228)

- Lines 215-216: The sentence should be rewritten to clarify.

Response: We have updated the manuscript accordingly.(Page 5, line 240-243)

- Line 247: Please add a S to play.

Response: We have updated the manuscript accordingly. (Page 6, line 273)

- Figure 2: Please correct for "inhibition of tubulin polymerization" and add a S to "function" in the title. Discard "The" before Rho and ROCK. Add a S to "microfilament". 

Response: We have updated the manuscript accordingly.(Page 10, line 321-326)

Round 2

Reviewer 2 Report

The review by Guangzhao Guan et al. has been improved by the modifications. Nevertheless, some concerns are still present and necessitate further improvement.
The new paragraph concerning effectors is quite complete, but has no logic. In fact, there are several categories of effectors (likewise for Ras) including kinases, regulators of actin polymerization and scaffolding proteins. It should be better positioned in this paragraph is such sorting has been used. In addition, most of the effectors cited are Rac1 or Cdc42 effectors, and RhoA effectors are (exception of Dia) not cited. Could it be more precise on the different GTPases also?
L235-237: The authors cite only one paper concerning the differences between ROCK1 and 2. There are other publications showing differences (examples: Rochelle et al., FASEB J, 2013 in which ROCK 1 specifically modulates Destrin (paralog of Cofilin) and not Cofilin; Wang et al., BMC Cancer, 2014 in which ROCK 1 is responsible for cofilin phosphorylation while ROCK 2 is not, contradicting the conclusions raised by Shi et al., cited here). In addition, differences in dimerization are present for ROCK1 and 2 that should be discussed. I suggest to develop this part further.
L290-292: The sentence doesn’t correspond to ROCK inhibitors and should then be suppressed or displaced.
L323: In the figure legend I found another MLC2. Please correct.

Author Response

10th Jan 2023

Genes

Re: Manuscript ID: genes-2140180-ori

We are grateful for reviewer 2’s comments. We have revised the manuscript based on the comments. All changes have been tracked in red in this revision.

Comments from Reviewer 2

The review by Guangzhao Guan et al. has been improved by the modifications. Nevertheless, some concerns are still present and necessitate further improvement.

The new paragraph concerning effectors is quite complete, but has no logic. In fact, there are several categories of effectors (likewise for Ras) including kinases, regulators of actin polymerization and scaffolding proteins. It should be better positioned in this paragraph is such sorting has been used. In addition, most of the effectors cited are Rac1 or Cdc42 effectors, and RhoA effectors are (exception of Dia) not cited. Could it be more precise on the different GTPases also?

Response: We have updated the review accordingly. (Page 5, line 204-217).

The effectors of three members of the family (RhoA, Rac1and Cdc42) have been well studied. RhoA regulates the contraction of moving cells, whereas Rac1 and Cdc42 mediate the formation of lamellipodia and filopodia, respectively [103]. Rho GTPase effectors can be classified into different groups according to their functions, such as scaffold proteins, kinases, actin binding proteins, formin-like molecules, and phospholipases [104]. For example, ROCK1,2 (serine/threonine kinases) and Dia1,2 (scaffold proteins) are two important RhoA effectors that regulate cytoskeleton nucleation and polymerization [95]. Wiskott–Aldrich syndrome protein (WASP; a scaffold protein) and neural-WASP (scaffold protein N-Wasp) are Cdc42 effectors. N-Wasp mediates actin polymerization via the Arp2/3 complex [105]. Formin-homology-domain-containing protein (formin-like molecule Fhod1) is one of the Rac1 effectors that regulates actin cytoskeleton organization and gene transcription [106]. p21-activated (serine/threonine) kinase is the effector for both Rac1 and Cdc42 [107]. It regulates cell shape and polarity through phosphorylation of multiple cytoskeletal proteins.”

L235-237: The authors cite only one paper concerning the differences between ROCK1 and 2. There are other publications showing differences (examples: Rochelle et al., FASEB J, 2013 in which ROCK 1 specifically modulates Destrin (paralog of Cofilin) and not Cofilin; Wang et al., BMC Cancer, 2014 in which ROCK 1 is responsible for cofilin phosphorylation while ROCK 2 is not, contradicting the conclusions raised by Shi et al., cited here).

Response: We have updated the review accordingly.(Page 5, line 243-248)

ROCK 1 is thought to be involved in destabilizing actin via regulating MLC and actin-myosin contraction, whereas ROCK 2 acts via cofilin to stabilize the actin cytoskeleton [122]. However, Wang et al., found that ROCK 1 phosphorylates LIM-kinase to inactivate cofilin, so regulating cell adhesion and invasion, but not ROCK 2 does not [123]. Also, Rochelle et al., demonstrated that ROCK 1 mediated amoeboid motility via destrin but not via cofilin [124].”

In addition, differences in dimerization are present for ROCK1 and 2 that should be discussed. I suggest to develop this part further.

This has been addressed on Page 7 (line 306-319).

Although ROCK 1 and ROCK 2 are highly homologous kinases and are both involved in the Rho/ROCK signaling pathway, their target substrates and physiological/pathological activities differ. Unfortunately, most ROCK inhibitors cannot selectively target one or other of the two isoforms; such inhibitors would be very useful. Dimerization is required for the RhoE activation of ROCK 1 [165], and although the homology between the kinase domains of ROCK 1 and ROCK 2 is ~90%, the homology between the N-terminal dimerization domains [166] of the two proteins is only ~60% [167]. Therefore dimerization-disrupting peptides could be developed and used to selectively target the dimerization domains of the two isoforms [167]. Autophosphorylation of ROCK 1 at Ser1333 and ROCK 2 at Ser1366 is required for activation of the kinases [168]. It has been shown that the interaction between Thr405 in the hydrophobic motif of ROCK 2 and Asp39 in the N-terminal extension was essential for both kinase activation and dimerization [169]. This provides further support for the development of peptides that prevent dimerization and kinase activity of ROCK isoforms.

L290-292: The sentence doesn’t correspond to ROCK inhibitors and should then be suppressed or displaced.

Response: We have removed this sentence. (Page 7, line 315)

L323: In the figure legend I found another MLC2. Please correct.

Response: We have updated the review accordingly. (Page 11, line 347)